analytical chemistry/green chemistry/spectroscopy

cefepime, cefazolin, fluorescamine, acriflavine, fluorimetry, pharmaceuticals

**Author for correspondence:**
Heba Abdel-Aziz
e-mail: dr.heba.abdelaziz11@gmail.com

This article has been edited by the Royal Society of Chemistry, including the commissioning, peer review process and editorial aspects up to the point of acceptance.

# Green and sensitive spectrofluorimetric method for the determination of two cephalosporins in dosage forms

Heba Abdel-Aziz, M. M. Tolba, N. El-Enany, F. A. Aly and M. E. Fathy

Department of Pharmaceutical Analytical Chemistry, Faculty of Pharmacy, Mansoura University, 35516 Mansoura, Egypt

HE-A, 0000-0003-1970-0245

Using two green and sensitive spectrofluorimetric methods, we quantified two cephalosporins, cefepime (CFM) and cefazolin (CFZ), in raw and pharmaceutical formulations. The first method is based on the reaction between CFM and fluorescamine (borate buffer, pH 8), which yields a highly fluorescent product. After excitation at 384 nm, the fluorescent product emits light at 484 nm. At concentration ranges from 12.0 to 120.0 ng ml$^{-1}$, the relative fluorescence intensity/concentration curve was linear with a limit of quantification (LOQ) of 2.46 ng ml$^{-1}$. The second method relied on measuring the CFZ quenching action on acriflavine fluorescence through formation of an ion-associate complex using Britton–Robinson buffer at pH 8. We measured acriflavine fluorescence at 505 nm after excitation at 265 nm. The decrease in acriflavine fluorescence intensity was CFZ concentration-dependent. Using this method, we quantified CFZ in concentrations ranging from 1 to 10 µg ml$^{-1}$ with a LOQ of 0.48 µg ml$^{-1}$. We studied and optimized the factors influencing reaction product formation. Moreover, we adapted our methods to the investigation of the mentioned drugs in raw and pharmaceutical formulations with greatly satisfying results. We statistically validated our methods according to International Council on Harmonisation Guidelines. The obtained results were consistent with those obtained with the official high-performance liquid chromatography methods.

# 1. Introduction

Cephalosporins are the most beneficial β-lactam antibiotics after penicillin. They are commonly used to treat bacterial infections [1]. Cephalosporins are semisynthetic antimicrobials derived from natural antibacterial, cephalosporin C, which was produced by the mould, *Cephalosporium acremonium*. Cephalosporins exert their bactericidal effect by inhibiting bacterial cell wall synthesis. Cephalosporins are usually classified by 'generation'. Cephalosporins from each generation usually have similar antibacterial activity, but it may partly depend on when they were discovered [1].

Cefepime (CFM) hydrochloride (figure 1*a*) is a broad-spectrum, fourth-generation parenteral cephalosporin used against pneumonia caused by many organisms and for urinary tract infections [1].

Cefazolin (CFZ) sodium (figure 1*b*) is a first-generation cephalosporin with wide-spectrum action against Gram-positive and Gram-negative bacteria [1].

The British Pharmacopoeia [2] and the United States Pharmacopeia [3] listed CFM and CFZ as standard medicinal products.

The existing quantification methods for these two cephalosporins include spectrophotometry [4–13], spectrofluorimetry after degradation [14], spectrofluorimetry through derivatization applying Hantzsch reaction [15], through derivatization with terbium [16] and derivatization with safranin [17]. Some reports also included high-performance liquid chromatography (HPLC) [18–20] and electrochemical [21–26] methods.

Since both CFM and CFZ have no native fluorescence, quantifying them through spectrofluorometry requires derivatization. In this study, we selected spectrofluorimetry for its high sensitivity, low cost and wide availability in most quality control laboratories. Besides their high sensitivity, our methods have several advantages over the previously published spectrofluorimetric methods. They are simple, rapid, inexpensive and environment-friendly. On the other hand, the previously published spectrofluorimetric methods suffered from using tedious and complex procedure with limited sensitivity as in the case of Hantzsch reaction [15], complexation with terbium which used highly expensive reagents [16] and reactions with safranin required toxic and environmentally harmful organic solvents [17]. Since we used water (a highly green solvent) as a solubilizing and diluting solvent, we considered our proposed quantification methods green. We aimed to create highly sensitive, simple, inexpensive and safe spectrofluorimetric methods to quantify the previously mentioned cephalosporins. In Method I, CFM reacted with fluorescamine in a borate buffer solution (pH 8) to form a product with a high emission at 484 nm after excitation at 384 nm. By contrast, Method II used the quenching impact of CFZ on acriflavine reagent native fluorescence. CFZ formed a non-fluorescent ion-associate complex with acriflavine in a Britton–Robinson buffer solution (pH 8). Moreover, Method II could assess the stability of CFZ since it depended on the presence of the carboxylic group that was absent in the degradation product of the drug [27].

# 2. Experimental set-up

## 2.1. Instrument

— We recorded the fluorescence spectra using a Cary Eclipse Spectrofluorometer fitted with Xenon flash lamp (Agilent Technology) with 5 nm slit width, using 1 cm quartz cells. For all experiments, we applied a high voltage of 800 V and a smoothing of 20. We adjusted the excitation and emission wavelengths at 384 and 484 nm for CFM and 265 and 505 nm for CFZ, respectively.
— We measured the pH values of buffer solutions using a Consort P-901 pH meter.

## 2.2. Materials and reagents

— We used analytical grade chemicals and spectroscopic grade solvents. We obtained the celphalosporins as kind gifts from their respective manufacturing companies: CFM from Chem Impex International (Wood Dale, IL), with a purity of $100.08 \pm 1.37\%$ as obtained using the official HPLC method, and CFZ from Bristol Myers Squibb (Cairo, Egypt), with a purity of $100.43 \pm 1.59\%$ as obtained using the official HPLC method. CFM vials (Batch no. B 25801) labelled to contain 500.0 mg CFM for injection; manufactured by Pharco B international company. Ziol vials (Batch

(a)

(b)

**Figure 1.** Structural formula of (a) cefepime and (b) cefazolin, where: (a): 1-[(6R,7R)-7-[2-(2-amino-4-thiazolyl)-glyoxylamido]-2-carboxy-8-oxo-5-thia-1-azabicyclo[4.2.0]oct-2-en-3-yl]methyl]-1 methylpyrrolidiniumchloride,72-(Z)-(O-methyloxime. (b): (6R,7R)-3-(((5-methyl-1,3,4-thiadiazol-2-yl)thio)methyl)-8-oxo-7-(2-(1H-tetrazol-1-yl)acetamidol)-5-thia-1 azabicyclo(4.2.0)oct-2-ene-2-carboxylate.

no. B 19422) labelled to contain 500.0 mg CFZ for injection; manufactured by Pharco B international company.

— We bought fluorescamine and acetone from Sigma, USA, and prepared a fluorescamine stock solution of 0.02% (w/v) in acetone. We prepared the 0.02 M borate buffer solution (pH 8) by combining adequate volumes of 0.02 M boric acid and sodium hydroxide.

— We prepared the $2 \times 10^{-6}$ mol l$^{-1}$ acriflavine solution (Sigma-Aldrich, USA) by transferring 2.5 ml of a $2 \times 10^{-4}$ mol l$^{-1}$ stock acriflavine solution (prepared by dissolving 0.013 g of acriflavine in 250 ml of double-distilled water) into a 250 ml volumetric flask and then filling the flask with double-distilled water.

— We prepared the Britton–Robinson buffer solution by mixing 0.02 M boric acid (Sigma-Aldrich, USA), 0.02 M phosphoric acid (Merck, Germany) and 0.02 M glacial acetic acid in equal volumes and adjusting the pH (2–12) using 0.02 M sodium hydroxide. We bought glacial acetic acid and sodium hydroxide from El-Nasr Pharmaceutical Chemicals Company, Egypt.

## 2.3. Preparation of stock solutions

We freshly prepared the CFM hydrochloride and CFZ sodium aqueous stock solutions by dissolving 10 mg of each cephalosporin separately in a 100 ml volumetric flask using double-distilled water to attain 100 µg ml$^{-1}$. We prepared the working solutions by diluting the stock solutions adequately using the same solvent.

## 2.4. Recommended procedures

### 2.4.1. Procedure for calibration graphs

For method I, we transferred accurately measured volumes of the CFM working solution into 10 ml volumetric flasks to obtain concentrations of 12–120 ng ml$^{-1}$. We added 1 ml of borate buffer (pH 8), mixed, added 1 ml of fluorescamine 0.02% (w/v), mixed and filled the flasks with double-distilled water to the mark. After excitation at 384 nm, we measured the fluorescence of the resulting solutions at 484 nm. We plotted the relative fluorescence intensity against CFM concentration and generated the corresponding regression equation.

For Method II, we transferred accurately measured volumes of the CFZ working solution into 10 ml volumetric flasks to obtain concentrations of 1–10 µg ml$^{-1}$. We added 1 ml of Britton–Robinson buffer (pH 8) and 0.9 ml of $2 \times 10^{-6}$ M acriflavine solution and then thoroughly mixed and filled the flasks with double-distilled water. After excitation at 265 nm, we measured the decrease in fluorescence intensity at 505 nm. To establish the calibration curve, we plotted the decrease in fluorescence intensity ($\Delta F$) against the CFZ concentration and then derived the regression equation. In both methods, we measured a blank sample (without the drug).

### 2.4.2. Procedure for assay of vials

We accurately weighed 10 mg of sterile powder from the CFM or zinol vials and transferred it to a 100 ml volumetric flask. We filled the flask with double-distilled water and sonicated it for 5 min. We then

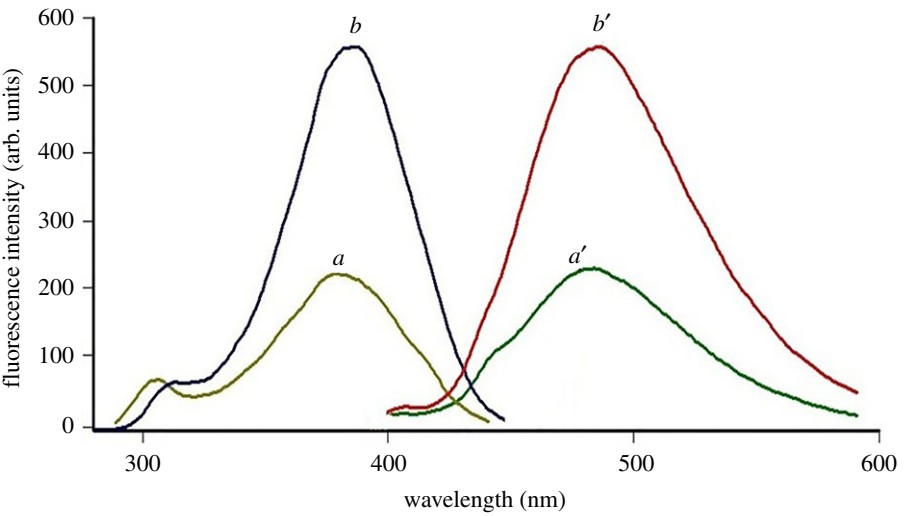

**Figure 2.** Fluorescence spectra of: ($a,a'$) blank fluorescamine at pH 8.0; ($b,b'$) cefepime (100 ng ml$^{-1}$) with fluorescamine at pH 8.0; ($a,b$) excitation spectra; ($a',b'$) emission spectra.

carried out the same procedure as stated under calibration graphs and calculated the concentrations using the derived regression equations.

# 3. Results and discussion

— Method I (using fluorescamine)

Fluorescamine was previously used to quantify many pharmaceutically important compounds with a primary amino group [28–31]. Fluorescamine is an extremely poor fluorescent derivatizing agent. However, it reacts swiftly with primary amines yielding derivatives with high fluorescence. Then, excess fluorescamine reacts with water to yield a non-fluorescent compound. It is thus useful to quantify water-soluble compounds with amine groups [32,33]. CFM, non-fluorescent compound with a primary amine, reacts with fluorescamine in a pH 8 borate buffer resulting in a fluorescent product. The formed fluorophore emits light at 484 nm after excitation at 384 nm (figure 2).

— Method II (using acriflavine)

Acridine dyes like acriflavine are highly fluorescent natural compounds that can be used as derivatizing agents [34]. Acriflavine was used for estimation of compounds having pharmaceutical interest such as ascorbic acid [35]. CFZ is non-fluorescent and has a free carboxylic group that ionizes (forming an anion) under alkaline conditions (pH 8). Under the same conditions, the cationic form of acriflavine predominates. The two compounds form a complex via electrostatic interaction between the two oppositely charged ions. The acriflavine reagent fluorescence intensity was measured at emission wavelength 505 nm subsequent to excitation at 265 nm and decreased appreciably upon addition of CFZ drug (figure 3). The quenching effect of CFZ was concentration-dependent (figure 4).

We then carefully investigated and optimized the experimental parameters that affected the reaction product for both methods.

Comparison was performed between the two suggested methods and the previously reported ones (table 1).

## 3.1. Experimental condition optimization

### 3.1.1. Method I

#### 3.1.1.1. Effect of pH

The fluorescence of the CFM-fluorescamine derivatization product occurs under alkaline conditions and disappeared under acidic conditions [31]. We thus performed our experiments using borate buffer at pH

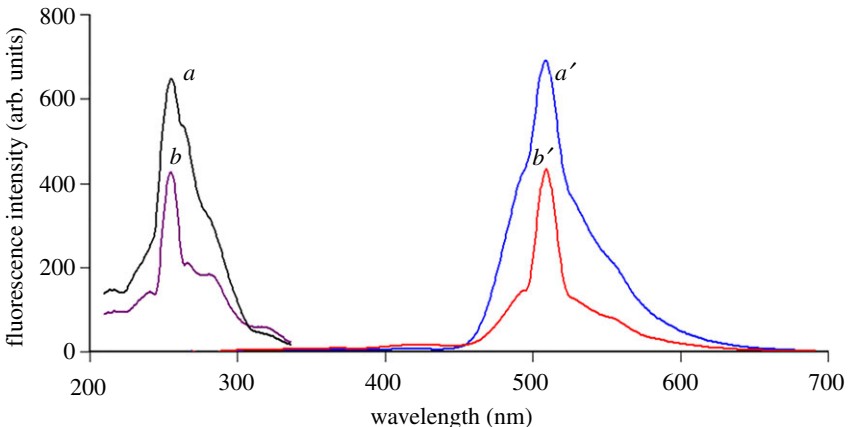

**Figure 3.** Fluorescence excitation and emission spectra. (*a*) Acriflavine ($2 \times 10^{-6}$ mol $I^{-1}$), pH 8. (*b*) Acriflavine ($2 \times 10^{-6}$ mol $I^{-1}$) with 8 μg ml$^{-1}$ cefazolin, pH 8.

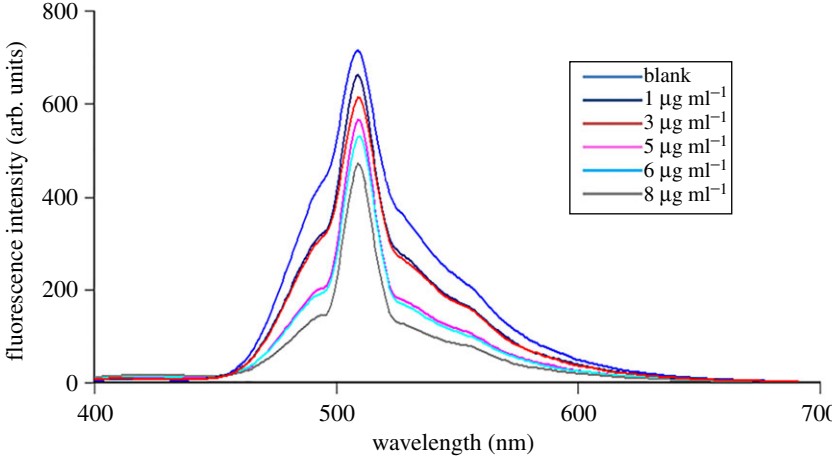

**Figure 4.** Fluorescence emission spectra of acriflavine with different concentrations of cefazolin (1.0, 3.0, 5.0, 6.0 and 8.0 μg ml$^{-1}$), pH 8.

7.5–12. We selected the borate buffer as it results in higher fluorescence intensity than other buffers of the same pH [31]. We obtained the highest fluorescence intensity at pH 8 ± 0.2 (figure 5*a*).

### 3.1.1.2. Effect of fluorescamine concentration
We assessed the impact of fluorescamine amount on the fluorescence intensity by recording the fluorescence intensity of solutions containing a fixed CFM concentration and different reagent volumes. We found that 1 ml of fluorescamine 0.02% (w/v) produced maximum fluorescence. At higher volumes, we observed a slight fluorescence intensity decrease (figure 6*a*).

### 3.1.1.3. Effect of reaction time and stability
We noticed that, at room temperature, the fluorescent product formed instantly, and the fluorescence intensity reached a maximum within 5 min. It remained stable for at least 3 h and then slowly decreased.

### 3.1.2. Method II

### 3.1.2.1. Effect of pH
We performed the CFZ-acriflavine reaction with the Britton–Robinson buffer at various pH (2–12) and observed the optimum quenching of fluorescence intensity with 1 ml of buffer at pH 8 (figure 5*b*).

### 3.1.2.2. Effect of acriflavine concentration
We obtained maximum fluorescence with 0.9 ml of $2 \times 10^{-6}$ mol l$^{-1}$ acriflavine (figure 6*b*).

**Table 1.** Comparison of the suggested methods with some reported ones.

| method | linearity | LOD | simplicity and materials |
|---|---|---|---|
| for cefepime | | | |
| spectrophotometry [4] | 2.0–24 µg ml$^{-1}$ | 0.41 µg ml$^{-1}$ | Harmful and time consuming as chloroform was used for extraction of the formed ion-associate complex. |
| spectrofluorimetry [14] | acid degradation 0.3–3.0 µg ml$^{-1}$ alkaline degradation 0.08–0.8 µg ml$^{-1}$ | 0.03 µg ml 0.01 µg ml$^{-1}$ | Tedious and time consuming as it needs heating at high temperature for 1.5 h with HCL and 1 h with NaOH and also needs time for pH neutralization. |
| spectrofluorimetry [15] | 1.0–50 µg ml$^{-1}$ | 0.24 µg ml$^{-1}$ | Tedious and time consuming as it needs boiling for 50 min. Harmful due to using acetyl acetone and formaldehyde. |
| spectrofluorimetry [17] | 0.15–1.35 µg ml$^{-1}$ | 40 ng ml$^{-1}$ | Using large amount of organic solvent (chloroform) has a very harmful and toxic effect on the analyst and environment. Complicated and time consuming in extraction steps. |
| HPLC [18] | 10–70 µg ml$^{-1}$ | n.a. | Time consuming. |
| the proposed method for CFM | 12.0–120.0 ng ml$^{-1}$ | 0.811 ng ml$^{-1}$ | Simple, rapid, inexpensive and environment-friendly. |
| for cefazolin | | | |
| spectrophotometry [7] | 20–80 µg ml$^{-1}$ | 1.58 µg ml$^{-1}$ | Tedious and time consuming as it needs heating in boiling water bath for 30 min. The resulted vapours are very irritant to eye. |
| spectrophotometry [10] | 40–360 µg ml$^{-1}$ | 9.39 µg ml$^{-1}$ | P-chloranilic acid is irritant reagent. |
| spectrofluorimetry [16] | $8.79 \times 10^{-6}$–$7.91 \times 10^{-5}$ M | $1.39 \times 10^{-6}$ M | Terbium oxide (Tb$_4$O$_7$) is a highly expensive reagent. Tedious and time consuming as preparation of stock solution of Tb$^{+3}$ needs to dissolve the reagent in nitric acid and evaporating the solution till dryness, then the residue was dissolved in 50 ml of 2 M hydrochloric acid, then dilution with water to prepare working solution. |
| the proposed method for CFZ | 1.0–10.0 µg ml$^{-1}$ | 0.159 µg ml$^{-1}$ | Effortless, fast, inexpensive and environment-friendly. |

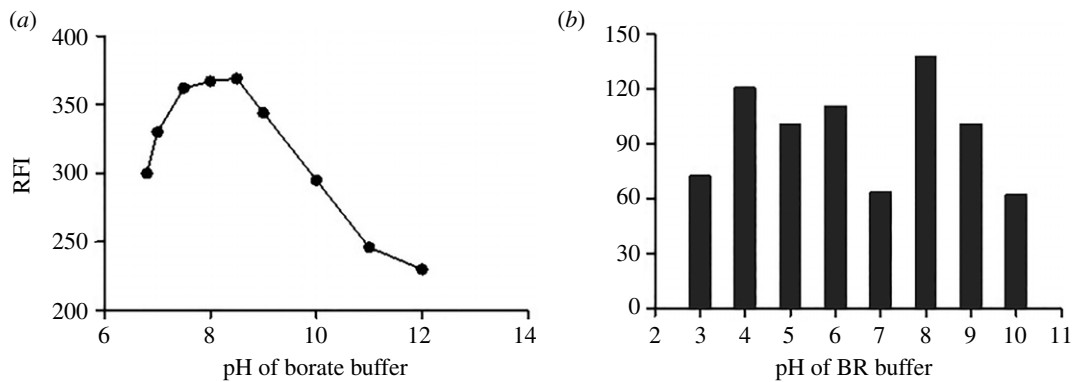

**Figure 5.** Effect of pH on the relative fluorescence intensity of the reaction product of (*a*) cefepime (100 ng ml$^{-1}$) with fluorescamine and (*b*) cefazolin (5 µg ml$^{-1}$) with acriflavine (2 × 10$^{-6}$ mol l$^{-1}$).

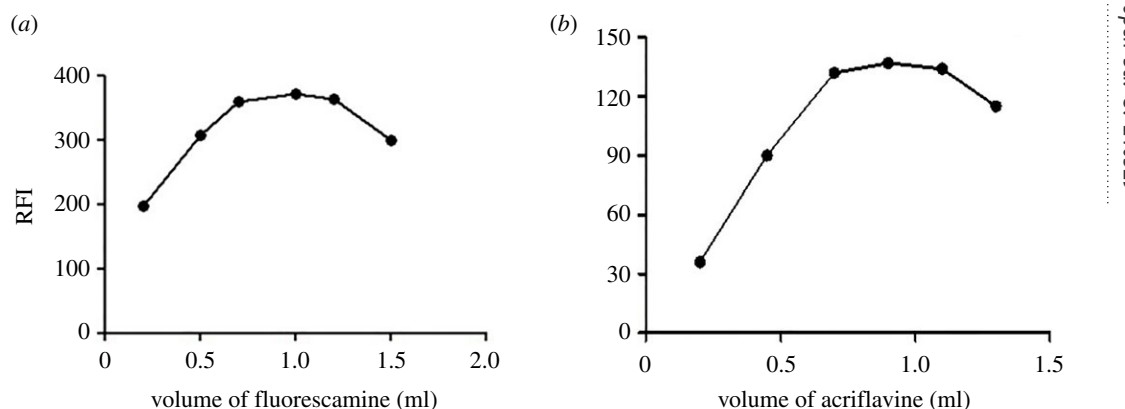

**Figure 6.** (*a*) Effect of volume of fluorescamine on the relative fluorescence intensity of the reaction product of cefepime (100 ng ml$^{-1}$) with fluorescamine at pH 8.0. (*b*) Effect of volume of acriflavine (2 × 10$^{-6}$ M) on the relative fluorescence intensity of the reaction product of acriflavine with cefazolin (5 µg ml$^{-1}$) at pH 8.0.

### 3.1.2.3. Effect of reaction time

At room temperature, the effect of reaction time has been studied to detect the time needed for complex formation which is indicated by maximum fluorescence quenching. The derivatization reaction occurred immediately after mixing and remained stable for up to 2 h.

## 3.2. Validation of the developed methods

To assess the validity of the two proposed methods, we determined the linearity, detection and quantification limits, accuracy, precision and specificity, as recommended by International Conference on Harmonization (ICH) Q2(R1) guidelines [36].

### 3.2.1. Linearity and quantification/detection limits

Under the optimal reaction conditions, the two proposed methods exhibit linearity in the ranges cited in table 2, with $r = 0.9999$. We calculated the limit of quantification (LOQ) and limit of detection (LOD) following the ICH Q2(R1) [36] using the following equations:

$$\text{LOQ} = 10\frac{S_a}{b} \quad \text{and} \quad \text{LOD} = 3.3\frac{S_a}{b},$$

where $S_a$ is the standard deviation of the calibration graph intercept and $b$ is the calibration graph slope. Table 2 shows the LOD and LOQ values for the investigated drugs.

**Table 2.** Analytical performance data for determination of cefepime and cefazolin using the two proposed methods.

| validation parameter | Method I using fluorescamine | Method II using acriflavine |
|---|---|---|
| $\lambda_{max}$, wavelength (nm) | 384/484 | 265/505 |
| linearity range | 12.0–120.0 ng ml$^{-1}$ | 1.0–10.0 µg ml$^{-1}$ |
| intercept ($a$) | −8.86 | 8.19 |
| slope ($b$) | 3.76 | 31.45 |
| correlation coefficient ($r$) | 0.9999 | 0.9999 |
| s.d. of residuals ($S_{y/x}$) | 1.202 | 1.772 |
| s.d. of intercept ($S_a$) | 0.925 | 1.516 |
| s.d. of slope ($S_b$) | 0.012 | 0.242 |
| limit of detection, LOD | 0.811 ng ml$^{-1}$ | 0.159 µg ml$^{-1}$ |
| limit of quantitation, LOQ | 2.458 ng ml$^{-1}$ | 0.482 µg ml$^{-1}$ |

**Table 3.** Determination of cefepime and cefazolin in their pure forms using the proposed spectrofluorimetric methods.

| pharmaceutical dosage forms | proposed spectrofluorimetric method (fluorescamine method) | | | reference methods [3] |
| | conc. taken (ng ml$^{-1}$) | conc. found (ng ml$^{-1}$) | % found[a] | % found[a] |
|---|---|---|---|---|
| cefepime | 12.0 | 12.190 | 101.59 | 99.88 |
| | 20.0 | 20.164 | 100.82 | 101.27 |
| | 50.0 | 49.667 | 99.34 | 98.22 |
| | 80.0 | 79.702 | 99.63 | 100.94 |
| | 100.0 | 99.902 | 99.90 | |
| | 120.0 | 120.369 | 100.31 | |
| mean ± s.d. | | 100.27 ± 0.83 | | 100.08 ± 1.37 |
| $t$ | | 0.27 (2.31)[b] | | |
| $F$ | | 2.73 (5.41)[b] | | |

| | proposed quenching method (acriflavine method) | | | |
| | conc. taken (ng ml$^{-1}$) | conc. found (ng ml$^{-1}$) | % found[a] | % found[a] |
|---|---|---|---|---|
| cefazolin | 1.0 | 1.011 | 101.12 | 101.93 |
| | 3.0 | 3.046 | 101.54 | 101.22 |
| | 5.0 | 4.985 | 99.72 | 98.24 |
| | 6.0 | 5.908 | 98.47 | 100.33 |
| | 8.0 | 8.006 | 100.08 | |
| | 10.0 | 10.041 | 100.42 | |
| mean ± s.d. | 100.23 ± 1.09 | | | 100.43 ± 1.60 |
| $t$ | | 0.24 (2.31)[b] | | |
| $F$ | | 2.16 (5.41)[b] | | |

[a]Average of three replicate determinations.
[b]The values between parentheses are the tabulated values of $t$ and $F$ at $p = 0.05$ [37].

### 3.2.2. Accuracy

Our methods are suitable to quantify the studied drugs over the concentration ranges shown in table 2. We compared the results of our methods with findings of the official chromatographic methods [3]. The

**Table 4.** Precision data for the determination of the the studied drugs using the two proposed spectrofluorimetric methods.

| | | intra-day precision | | | inter-day precision | | |
|---|---|---|---|---|---|---|---|
| | | mean ± s.d. | RSD (%) | % error | mean ± s.d. | RSD (%) | % error |
| concentration (ng ml$^{-1}$) | | | | | | | |
| Method I for CFM | 20.0 | 100.48 ± 0.65 | 0.65 | 0.38 | 99.91 ± 0.91 | 0.91 | 0.53 |
| | 80.0 | 99.97 ± 0.36 | 0.36 | 0.21 | 99.65 ± 1.06 | 1.06 | 0.61 |
| | 100.0 | 100.43 ± 0.55 | 0.55 | 0.32 | 100.52 ± 0.56 | 0.56 | 0.32 |
| concentration (µg ml$^{-1}$) | | | | | | | |
| Method II for CFZ | 3.0 | 100.55 ± 0.97 | 0.97 | 0.56 | 100.35 ± 1.07 | 1.07 | 0.62 |
| | 5.0 | 100.34 ± 1.01 | 1.00 | 0.58 | 100.07 ± 0.81 | 0.81 | 0.47 |
| | 8.0 | 100.07 ± 0.93 | 0.93 | 0.54 | 99.02 ± 1.03 | 1.04 | 0.60 |

**Table 5.** Application of the proposed spectrofluorimetric methods to the determination of cefepime and cefazolin in their own dosage forms.

| | proposed spectrofluorimetric method | | | reference methods [3] |
|---|---|---|---|---|
| pharmaceutical dosage forms | conc. taken (ng ml$^{-1}$) | conc. found (ng ml$^{-1}$) | % found[a] | % found[a] |
| cefepime vials CFM | 12.0 | 12.146 | 101.22 | 99.88 |
| (500 mg) | 50.0 | 49.905 | 99.81 | 101.27 |
| | 100.0 | 99.546 | 99.55 | 98.22 |
| | 120.0 | 120.406 | 100.34 | 100.94 |
| mean | | | 100.23 | 100.08 |
| ±s.d. | | | 0.74 | 1.37 |
| %RSD | | | 0.73 | |
| t | | | 0.19 (2.45)[b] | |
| F | | | 3.47 (9.28)[b] | |
| | proposed quenching method | | | |
| | conc. taken (µg ml$^{-1}$) | conc. found (µg ml$^{-1}$) | % found[a] | |
| zinol vials CFZ (500 mg) | 1.0 | 1.010 | 101.02 | 101.93 |
| | 5.0 | 4.956 | 99.14 | 101.22 |
| | 8.0 | 8.062 | 100.78 | 98.24 |
| | 10.0 | 9.971 | 99.71 | 100.33 |
| mean | | | 100.16 | 100.43 |
| ±s.d. | | | 0.89 | 1.60 |
| %RSD | | | 0.88 | |
| t | | | 0.29 (2.45)[b] | |
| F | | | 3.24 (9.28)[b] | |

[a]Average of three replicate determinations.
[b]The values between parentheses are the tabulated values of t and F at $p = 0.05$ [37].

official CFM quantification method is based on HPLC using sodium-1-pentane sulfonate solution/ acetonitrile (94 : 6 v/v) as a mobile phase with UV detection at 254 nm. The official CFZ quantification method also uses chromatography with anhydrous dibasic sodium phosphate and citric acid solution/acetonitrile (90 : 10 v/v) as a mobile phase and UV detection at 254 nm. The statistical comparison [37] of our methods and the official methods [3] using Student's t-test and variance ratio F-test revealed no substantial accuracy and precision differences (table 3).

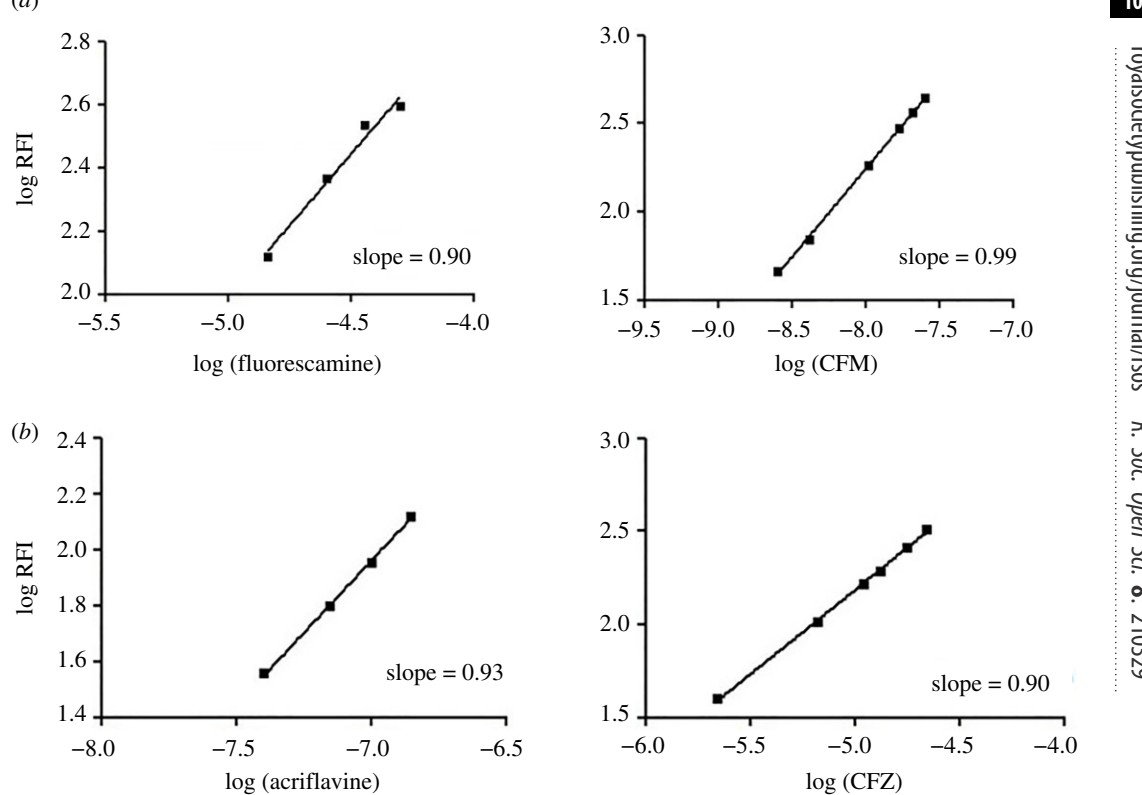

**Figure 7.** (*a*) Stoichiometry of the reaction between cefepime and fluorescamine using limiting logarithmic method. (*b*) Stoichiometry of the reaction between cefazolin and acriflavine using limiting logarithmic method.

### 3.2.3. Precision

To check precision, we used three concentrations and performed triplicate measurements for each of them. Table 4 shows the results. The % relative standard deviation (RSD) values were inferior to 2, which confirmed the acceptable precision of our methods.

### 3.2.4. Robustness

To assess the robustness of our methods, we measured fluorescence constancy with little intentional variations in the empirical conditions such as fluorescamine volume ($1.0 \pm 0.2$ ml) for method I, acriflavine volume ($0.9 \pm 0.2$ ml) for method II, and change in pH ($8.0 \pm 0.2$) for both methods. These minute alterations might occur during practical experiments and did not affect the emission intensity of the formed product.

### 3.2.5. Selectivity

Since the spectrofluorimetric quantification of the two cephalosporins was based on chemical reactions, our methods have high selectivity towards the investigated drugs due to the presence of certain functional groups. In Method II, CFZ can be quantified in the presence of its degradation product.

## 3.3. Application of the proposed methods to pharmaceuticals analysis

Our methods are suitable for the quantification of CFM and CFZ in their dosage forms. Our results showed good agreement with findings obtained using the official methods [3] (table 5). Student's *t*-test and *F*-test showed that there was no appreciable difference in their performances [37].

**Scheme 1.** Reaction mechanism of cefepime with fluorescamine.

**Scheme 2.** Mechanism for the formation of an ion-associated complex between cefazolin and acriflavine.

**Table 6.** Analytical eco-scale score for the two proposed spectrofluorimetric methods.

| | penalty points | |
| --- | --- | --- |
| | Method II | Method I |
| reagents | | |
| fluorescamine | 1 | |
| acetone (solvent for fluorescamine) | 4 | |
| borate buffer | 0 | |
| acriflavine | | 6 |
| BRB | | 2 |
| item | | |
| spectrofluorimeter | 0 | |
| waste (1–10 ml) | 3 | |
| occupational hazard (no vapours or gases) | 0 | |
| total penalty points | $\sum$ 8 | $\sum$ 11 |
| analytical eco-scale score | 92 | 89 |

## 3.4. Reaction pathways

We studied the reaction stoichiometry using the slope–ratio method. Plotting log fluorescence intensity versus either log (fluorescamine) or log (CFM) resulted in straight lines with slopes of 0.90 and 0.99, respectively (figure 7a). Thus, the reaction progressed in a 1 : 1 ratio. Similarly, plotting log fluorescence intensity versus log (acriflavine) or log (CFZ) resulted in straight lines with slopes of 0.93 and 0.90; respectively (figure 7b). We concluded that the reaction took place in the ratio of 1 : 1. Using previous results, we proposed reaction pathways for each method (schemes 1 and 2).

Since the reaction of acriflavine with CFZ depends on the carboxylic group (which quenches the reagent fluorescence) and the CFZ degradation product is decarboxylated, method II would be suitable to assess the stability of this compound [27].

# 4. Assessment of greenness

We assessed the greenness of our methods using the analytical eco-scale method. Methods I (fluorescamine) and II (acriflavine) obtained scores of 92 and 89, respectively, which are both excellent green (table 6).

# 5. Conclusion

We applied two different simple derivatization reactions to quantify two important cephalosporins, CFM and CFZ, in their dosage forms. Our methods are rapid, sensitive, effortless and cheap. Moreover, these methods are green (they have little impact on the environment). Another important feature is their high selectively for the studied drugs, thanks to the selectivity of the reactions. Moreover, method II can assess the stability of CFZ because acriflavine does not react with the CFZ degradation product.

Data accessibility. The data of the work were deposited in the Dryad Digital Repository: https://doi.org/10.5061/dryad. 866t1g1q3 [38].

Authors' contributions. H.A.E. performed methodology, practical work, writing, reviewing, and drafting of the paper; M.M.T., M.E.F. carried out data curation, validation, writing, and editing; N.E.-E., F.A.A. coordinated the work and thoroughly revised the manuscript. All authors approved the final form of the manuscript to be published.

Competing interests. We declare we have no competing interests.

Funding. We received no funding for this study.

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
