## [Peer Review File · Royal Society Open Science]

Review History

RSOS-210329.R0 (Original submission)

Review form: Reviewer 1

Is the manuscript scientifically sound in its present form?

Yes

Are the interpretations and conclusions justified by the results?

Yes

Is the language acceptable?

Yes

Do you have any ethical concerns with this paper?

No

Have you any concerns about statistical analyses in this paper?

No

Recommendation?

Accept with minor revision (please list in comments)

Comments to the Author(s)

- 1- Please make a comparison to illustrate the advantages of the developed method with the previous methods regarding sensitivity, linearity, simplicity and materials used so on
- 2- Regarding the selectivity, have you checked the additives usually combined with CFM and CFZ? Please mention more details about the studied additives.
- 3- I think it is better to combine figures of effect of pH together, and figures of volume of fluorecamine and volume of acriflavine. Clearly combine figure 5 and 7 together and combine figure 6 and 9.
- 4- In figure 5: add pH of borate buffer instead of pH only.

Review form: Reviewer 2**Is the manuscript scientifically sound in its present form?**

Yes

Are the interpretations and conclusions justified by the results?

Yes

Is the language acceptable?

Yes

Do you have any ethical concerns with this paper?

Yes

Have you any concerns about statistical analyses in this paper?

Yes

Recommendation?

Major revision is needed (please make suggestions in comments)

Comments to the Author(s)

The manuscript describes two spectrofluorometric methods for the determination of two cephalosporins. . The paper in its present form is not suitable for publication. However, after major revision, it can be reconsidered for possible publication..The authors have to address the following points:

1. Title

I suggest to be more concise , I suggest

Green and Sensitive Spectrofluorometric method for the determination of two cephalosporins in dosage forms..

1. What are the pros and cons of the proposed method
2. What about the stability of the standard solutions.
3. Number of figures should be reduced, many figures can be deleted!!
4. Fig.#2&3; the blank has spectra exactly similar to the those of the fluorophore.
5. Table#2:
the F value (0.009) can nver be a fraction!!!!. Please re-calculate!!
the Heading of the table needs to be checked !

6. Table#1, Method #1:
 - the working range is 12-120 ug/ml
 - the Value of LOQ is 2.4 ug/ml
 - you can use lower range ,eg. 5 ug/ml
7. Scheme #2; it is better to be represented as two ion pairs with positive and negative charges.
8. Language needs to be polished.

Decision letter (RSOS-210329.R0)

Dear Dr Abdel-aziz:

Title: Green and Sensitive Spectrofluorimetric Determination of Two Pharmaceutically Important Cephalosporin Drugs in Their Dosage Forms
Manuscript ID: RSOS-210329

The editor assigned to your manuscript has now received comments from reviewers. We would like you to revise your paper in accordance with the referee and Subject Editor suggestions which can be found below (not including confidential reports to the Editor). Please note this decision does not guarantee eventual acceptance.

Please submit your revised paper before 07-May-2021. Please note that the revision deadline will expire at 00.00am on this date. If we do not hear from you within this time then it will be assumed that the paper has been withdrawn. In exceptional circumstances, extensions may be possible if agreed with the Editorial Office in advance. We do not allow multiple rounds of revision so we urge you to make every effort to fully address all of the comments at this stage. If deemed necessary by the Editors, your manuscript will be sent back to one or more of the original reviewers for assessment. If the original reviewers are not available we may invite new reviewers.

RSC Associate Editor:
Comments to the Author:
(There are no comments.)

RSC Subject Editor:
Comments to the Author:
(There are no comments.)

Reviewers' Comments to Author:

Reviewer: 1

Comments to the Author(s)

- 1- Please make a comparison to illustrate the advantages of the developed method with the previous methods regarding sensitivity, linearity, simplicity and materials used so on
- 2- Regarding the selectivity, have you checked the additives usually combined with CFM and CFZ? Please mention more details about the studied additives.
- 3- I think it is better to combine figures of effect of pH together, and figures of volume of fluorescamine and volume of acriflavine. Clearly combine figure 5 and 7 together and combine figure 6 and 9.
- 4- In figure 5: add pH of borate buffer instead of pH only.

Reviewer: 2

Comments to the Author(s)

The manuscript describes two spectrofluorometric methods for the determination of two cephalosporins. . The paper in its present form is not suitable for publication. However, after major revision, it can be reconsidered for possible publication..The authors have to address the following points:

1. Title

I suggest to be more concise , I suggest

Green and Sensitive Spectrofluorometric method for the determination of two cephalosporins in dosage forms..

1. What are the pros and cons of the proposed method
2. What about the stability of the standard solutions.
3. Number of figures should be reduced, many figures can be deleted!!

4. Fig.#2&3; the blank has spectra exactly similar to the those of the fluorophore.
5. Table#2:
 - the F value (0.009) can nver be a fraction!!!. Please re-calculate!!
 - the Heading of the table needs to be checked !
6. Table#1, Method #1:
 - the working range is 12-120 ug/ml
 - the Value of LOQ is 2.4 ug/ml
 - you can use lower range ,eg. 5 ug/ml
7. Scheme #2; it is better to be represented as two ion pairs with positive and negative charges.
8. Language needs to be polished.

Author's Response to Decision Letter for (RSOS-210329.R0)

See Appendix A.

RSOS-210329.R1 (Revision)

Review form: Reviewer 1

Is the manuscript scientifically sound in its present form?

Yes

Are the interpretations and conclusions justified by the results?

Yes

Is the language acceptable?

Yes

Do you have any ethical concerns with this paper?

No

Have you any concerns about statistical analyses in this paper?

No

Recommendation?

Accept as is

Comments to the Author(s)

There are few grammatical mistakes that should be rechecked.

Review form: Reviewer 2

Is the manuscript scientifically sound in its present form?

Yes

Are the interpretations and conclusions justified by the results?

Yes

Is the language acceptable?

Yes

Do you have any ethical concerns with this paper?

No

Have you any concerns about statistical analyses in this paper?

No

Recommendation?

Accept as is

Comments to the Author(s)

The manuscript after revision is suitable for publication

Decision letter (RSOS-210329.R1)

Dear Dr Abdel-aziz:

Title: Green and Sensitive Spectrofluorimetric Method for the Determination of Two Cephalosporins in Dosage Forms
Manuscript ID: RSOS-210329.R1

Thank you for submitting the above manuscript to Royal Society Open Science. On behalf of the Editors and the Royal Society of Chemistry, I am pleased to inform you that your manuscript will be accepted for publication in Royal Society Open Science subject to minor revision in accordance with the referee suggestions. Please find the reviewers' comments at the end of this email.

The reviewers and handling editors have recommended publication, but also suggest some minor revisions to your manuscript. Therefore, I invite you to respond to the comments and revise your manuscript.

Because the schedule for publication is very tight, it is a condition of publication that you submit the revised version of your manuscript before 04-Jun-2021. Please note that the revision deadline will expire at 00.00am on this date. If you do not think you will be able to meet this date please let me know immediately.

Kind regards,
Dr Laura Smith
Publishing Editor, Journals

RSC Associate Editor:
Comments to the Author:
According to the comment of one reviewer, the decision was made.

RSC Subject Editor:
Comments to the Author:
(There are no comments.)

Reviewer comments to Author:
Reviewer: 1
Comments to the Author(s)
There are few grammatical mistakes that should be rechecked.

Reviewer: 2
Comments to the Author(s)
The manuscript after revision is suitable for publication

Author's Response to Decision Letter for (RSOS-210329.R1)

See Appendices B & C.

Decision letter (RSOS-210329.R2)

Dear Dr Abdel-aziz:

Title: Green and Sensitive Spectrofluorimetric Method for the Determination of Two Cephalosporins in Dosage Forms
Manuscript ID: RSOS-210329.R2

It is a pleasure to accept your manuscript in its current form for publication in Royal Society Open Science. The chemistry content of Royal Society Open Science is published in collaboration with the Royal Society of Chemistry.

RSC Associate Editor
Comments to the Author:
(There are no comments.)

Reviewer(s)' Comments to Author:

Appendix A

List of changes

Thank you very much for considering our manuscript of importance. We are very grateful for your valuable comments. We fixed all the comments revised by the reviewers and we revised our manuscript again carefully. These comments and suggestions improved the quality of this manuscript. Also, we would like to discuss them again point by point. If any other comments needed, we will be grateful if you tell us again.

Reviewer #1:

1- Please make a comparison to illustrate the advantages of the developed method with the previous methods regarding sensitivity, linearity, simplicity and materials used so on

The comparison was done in table (1) in the manuscript.

2- Regarding the selectivity, have you checked the additives usually combined with CFM and CFZ? Please mention more details about the studied additives.

Dosage form is vial and vial has no additives. I have studied the common additives which present in dosage forms and it has no effect.

Selectivity that I mean is due to reaction selectivity to certain group.

3- I think it is better to combine figures of effect of pH together, and figures of volume of fluorecamine and volume of acriflavine. Clearly combine figure 5 and 7 together and combine figure 6 and 9.

Ok, combination of figures was performed.

4- In figure 5: add pH of borate buffer instead of pH only.

Ok, pH was modified to pH of borate buffer in figure 5.

Reviewer #2:

1-Title

I suggest to be more concise, I suggest

Green and Sensitive Spectrofluorometric method for the determination of two cephalosporins in dosage forms.

(Ok, there title was changed to your suggested one to be more concise).

1. What are the pros and cons of the proposed method?

I have made a table illustrating the advantages of the developed method with the previously reported ones (Table 1).

I did not find a real disadvantage of the two proposed methods since the methods are characterized by simplicity, sensitivity and green impact on analyst and environment.

2. What about the stability of the standard solutions.

Standard solutions were freshly prepared every day at analysis time as they are not stable for long time.

3. Number of figures should be reduced, many figures can be deleted!!

I have combined figure 5 and 7 together and figure 6 and 9 together.

Figure 8 was removed from the manuscript.

4. Fig.#2&3; the blank has spectra exactly similar to the those of the fluorophore.

Fig. 2: This is a constant noise from the blank (buffer solution) due to using high sensitivity voltage (800 V). This blank was considered during construction of the calibration curve.

Fluorescamine (fluorogenic agent) in borate buffer has RFI value as in case of previously published papers[1].

Fig. 3: The two spectra for acriflavine, A: for acriflavin alone and B: after quenching which mean that (acriflavin- CFZ) has no fluorescence it's not a fluorophore.

5. Table#2:

the F value (0.009) can never be a fraction!!!. Please re-calculate!!

The Heading of the table needs to be checked !

There was an error in the calculation of F value

Recalculation were done and the true value is 2.73

The table heading was changed to (Determination of Cefepime and cefazolin in their pure forms using the proposed spectrofluorimetric methods).

6. Table#1, Method #1:
the working range is 12-120 ug/ml
the Value of LOQ is 2.4 ug/ml
you can use lower range, eg. 5 ug/ml

We tried to include lower concentration, but it was non linear with the calibration curve.

7. Scheme #2; it is better to be represented as two ion pairs with positive and negative charges.

All published papers concerning ion associate complex of acriflavine with drugs showed the mechanism as expected in our manuscript [2-4]

8. Language needs to be polished.

The manuscript was subjected to language editing in Egyptian knowledge bank and their letter was previously sent to the journal.

Thank you very much for your valuable comments. If any other comments needed, please tell us.

Please accept our best regards.

References

1. Walsh, M., et al., *Simple and sensitive spectrofluorimetric method for the determination of pregabalin in capsules through derivatization with fluorescamine*. Luminescence, 2011. **26**(5): p. 342-348.
2. Yang, H., et al., *A sensitive fluorescence quenching method for the detection of tartrazine with acriflavine in soft drinks*. Luminescence, 2018. **33**(2): p. 349-355.
3. Ibrahim, F., H. Elmansi, and R. Aboshabana, *Assessment of two analgesic drugs through fluorescence quenching of acriflavine as a new green methodology*. Microchemical Journal, 2021. **164**: p. 105882.
4. Tolba, M. and H. Elmansi, *Studying the quenching resulted from the formation of an association complex between olsalazine or sulfasalazine with acriflavine*. Royal Society Open Science, 2021. **8**(4): p. 210110.

Appendix B

List of changes

Thank you very much for considering our manuscript of importance. We are very grateful for your valuable comments. We fixed all the comments revised by the reviewers and we revised our manuscript again carefully. These comments and suggestions improved the quality of this manuscript. Also, we would like to discuss them again point by point. If any other comments needed, we will be grateful if you tell us again.

Reviewer #1:

1- Please make a comparison to illustrate the advantages of the developed method with the previous methods regarding sensitivity, linearity, simplicity and materials used so on

The comparison was done in table (1) in the manuscript.

2- Regarding the selectivity, have you checked the additives usually combined with CFM and CFZ? Please mention more details about the studied additives.

Dosage form is vial and vial has no additives. I have studied the common additives which present in dosage forms and it has no effect.

Selectivity that I mean is due to reaction selectivity to certain group.

3- I think it is better to combine figures of effect of pH together, and figures of volume of fluorecamine and volume of acriflavine. Clearly combine figure 5 and 7 together and combine figure 6 and 9.

Ok, combination of figures was performed.

4- In figure 5: add pH of borate buffer instead of pH only.

Ok, pH was modified to pH of borate buffer in figure 5.

Reviewer #2:

1-Title

I suggest to be more concise, I suggest Green and Sensitive Spectrofluorometric method for the determination of two cephalosporins in dosage forms.

(Ok, there title was changed to your suggested one to be more concise).

1. What are the pros and cons of the proposed method?

I have made a table illustrating the advantages of the developed method with the previously reported ones (Table 1).

I did not find a real disadvantage of the two proposed methods since the methods are characterized by simplicity, sensitivity and green impact on analyst and environment.

2. What about the stability of the standard solutions.

Standard solutions were freshly prepared every day at analysis time as they are not stable for long time.

3. Number of figures should be reduced, many figures can be deleted!!

I have combined figure 5 and 7 together and figure 6 and 9 together.

Figure 8 was removed from the manuscript.

4. Fig.#2&3; the blank has spectra exactly similar to the those of the fluorophore.

Fig. 2: This is a constant noise from the blank (buffer solution) due to using high sensitivity voltage (800 V). This blank was considered during construction of the calibration curve.

Fluorescamine (fluorogenic agent) in borate buffer has RFI value as in case of previously published papers[1].

Fig. 3: The two spectra for acriflavine, A: for acriflavin alone and B: after quenching which mean that (acriflavin- CFZ) has no fluorescence it's not a fluorophore.

5. Table#2:

the F value (0.009) can never be a fraction!!!. Please re-calculate!!

The Heading of the table needs to be checked !

There was an error in the calculation of F value

Recalculation were done and the true value is 2.73

The table heading was changed to (Determination of Cefepime and cefazolin in their pure forms using the proposed spectrofluorimetric methods).

6. Table#1, Method #1:
the working range is 12-120 ug/ml
the Value of LOQ is 2.4 ug/ml
you can use lower range, eg. 5 ug/ml

We tried to include lower concentration, but it was non linear with the calibration curve.

7. Scheme #2; it is better to be represented as two ion pairs with positive and negative charges.

All published papers concerning ion associate complex of acriflavine with drugs showed the mechanism as expected in our manuscript [2-4]

8. Language needs to be polished.

The manuscript was subjected to language editing in Egyptian knowledge bank and their letter was previously sent to the journal.

Thank you very much for your valuable comments. If any other comments needed, please tell us.

Please accept our best regards.

References

1. Walsh, M., et al., *Simple and sensitive spectrofluorimetric method for the determination of pregabalin in capsules through derivatization with fluorescamine*. Luminescence, 2011. **26**(5): p. 342-348.
2. Yang, H., et al., *A sensitive fluorescence quenching method for the detection of tartrazine with acriflavine in soft drinks*. Luminescence, 2018. **33**(2): p. 349-355.
3. Ibrahim, F., H. Elmansi, and R. Aboshabana, *Assessment of two analgesic drugs through fluorescence quenching of acriflavine as a new green methodology*. Microchemical Journal, 2021. **164**: p. 105882.
4. Tolba, M. and H. Elmansi, *Studying the quenching resulted from the formation of an association complex between olsalazine or sulfasalazine with acriflavine*. Royal Society Open Science, 2021. **8**(4): p. 210110.

Appendix C

There were few grammatical mistakes that was checked and highlighted in yellow colour.

Thank you